# VALUE-AS-RETURN: A TWO-STAGE FRAMEWORK TO ALIGN ON THE OPTIMAL SCORE FUNCTION

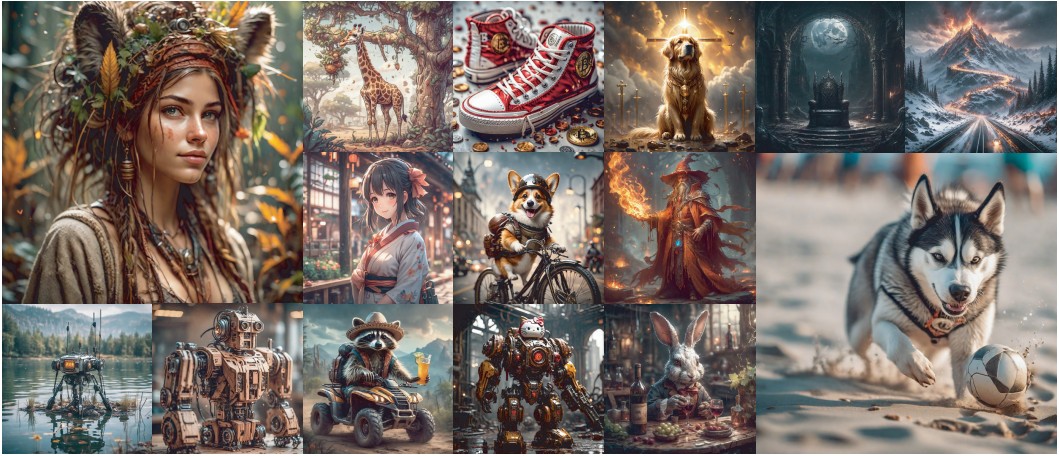

Figure 1: Generated samples of VRPO-SDXL. See Appendix I for the corresponding prompts.

## ABSTRACT

Reinforcement learning with diffusion models has shown strong potential, but existing approaches such as variants of Direct Preference Optimization (DPO) often rely on an inaccurate simplification: they equate trajectory likelihoods with final-state probabilities. This mismatch leads to suboptimal alignment. We address this limitation with a principled framework that leverages the optimal value function as the return for short trajectory segments. Our approach follows a two-stage procedure: (i) learning a value-distribution function to estimate segment-level returns, and (ii) applying our VRPO to refine the score function. We prove that, under sufficient model capacity, the resulting model is equivalent to training a diffusion process on the tilted distribution proportional to $p(x)\exp(\eta r(x))$. Experiments on large-scale diffusion models validate our analysis and show stable and consistent improvements over prior methods. **Anonymous** website for reproducibility: https://osf.io/7a4fh/?view_only=f951fa68c7ef43a19a44ee38b11847f6.

## 1 INTRODUCTION

Reinforcement learning(Wallace et al., 2023; Black et al., 2023; Agarwal et al., 2019) (RL) has become an increasingly important tool for guiding generative models, and diffusion models (Song et al., 2021a;b; Ho et al., 2020) in particular, toward producing outputs that are both high-quality and well-aligned with task-specific objectives. By incorporating reward signals, RL provides a way to capture complex preferences that are not easily expressed through likelihood-based training alone. Despite these advantages, applying RL to diffusion models remains challenging due to the difficulty of faithfully integrating rewards defined in the image domain into the trajectory-based training process. A prominent example is given by existing variants of Direct Preference Optimization (Wallace et al., 2023; Rafailov et al., 2023) (DPO), which commonly treat pairwise preferences over final states as if they directly reflected trajectory-level preferences. This oversimplification often misrepresents the underlying reward structure, leading to misaligned policies and degraded generation quality.

Several recent works (Yang et al., 2023; Liang et al., 2024) attempt to mitigate this issue by decomposing long diffusion trajectories into shorter segments. However, such approaches typically rely on heuristic design choices or assumptions that may not consistently hold in practice. As a result, they can introduce inconsistencies across time steps and undermine the stability of reward propagation, further complicating optimization and limiting generalization.

In this work, we propose a principled two-stage RL framework for diffusion models that is grounded in stochastic optimal control (Domingo-Enrich et al., 2025)(SOC). **The key idea is to leverage the optimal value function as the return signal for short trajectory segments**, thereby aligning training objectives more closely with the true reward structure. Our framework proceeds in two stages. In the first stage, we learn an optimal value-distribution function that characterizes the distribution of returns from intermediate states. In the second stage, we employ our stage-wise VRPO to refine the diffusion model, using preferences derived from the learned value function. This decomposition allows us to preserve reward consistency across segments while making long-horizon optimization tractable.

Our contributions are threefold:

- We introduce a two-stage RL framework that combines optimal value-distribution function training with our VRPO, providing a principled approach to align on the optimal score function.

- We prove that, under sufficient model capacity, our method converges to the optimal solution of training a diffusion process on the tilted distribution, which is proportional to $p(\mathbf{x}) \exp(\eta r(\mathbf{x}))$, where $r$ is the reward function in the image domain, and $p(\cdot)$ is the original distribution.

- We propose a CDF-based training strategy for the optimal value-distribution function, enhancing the training stability and robustness.

We validate our theoretical findings through extensive experiments on large-scale diffusion models(Rombach et al., 2022; Podell et al., 2023), demonstrating stable and consistent performance improvements across a variety of tasks.

## 2 PRELIMINARY

### 2.1 DIFFUSION GENERATION PROCESS AS CONTINUOUS TIME MARKOV CHAIN

To be clear, all the notions ignore the text condition $c$. Suppose the forward diffusion process is defined by the following SDE:

$$\mathrm{d}\mathbf{x}'_\tau = -\mathbf{f}(\mathbf{x}'_\tau, 1 - \tau)\,\mathrm{d}\tau + \sigma(1 - \tau)\,\mathrm{d}\mathbf{B}'_\tau, \qquad \tau \in [0, 1], \tag{1}$$

where $\mathbf{x}'_\tau \in \mathbb{R}^d$ is the state, $\mathbf{B}'_\tau$ is a $d$-dimensional standard Brownian motion, $\mathbf{f} : \mathbb{R}^d \times [0, 1] \to \mathbb{R}^d$ is the drift, and $\sigma : [0, 1] \to \mathbb{R}_+$ is the diffusion scale (e.g., VP/VE families). The process starts from $\mathbf{x}'_0 \sim p_{\text{data}}$ (data distribution), and ends at $\mathbf{x}'_1 \sim \mathcal{N}(\mathbf{0}, \mathbf{I})$.

Then, the reverse-time SDE is given by (Anderson, 1982; Song et al., 2021b):

$$\mathrm{d}\mathbf{x}_t = \left(\mathbf{f}(\mathbf{x}_t, t) + \sigma(t)^2 \nabla_\mathbf{x} \log p_{\text{ref}}(\mathbf{x}_t, t)\right)\mathrm{d}t + \sigma(t)\,\mathrm{d}\mathbf{B}_t, \qquad t \in [0, 1], \tag{2}$$

where $\mathbf{B}_t$ is a $d$-dimensional standard Brownian motion, $p_{\text{ref}}(\mathbf{x}, t)$ denotes the (reference/base) marginal density of $\mathbf{x}_t$ at time $t$. Note that $\mathbf{x}_t$ and $\mathbf{x}'_{1-t}$ follow the same distribution. We use boldface $\mathbf{x}$ for states and write $\boldsymbol{x}_{0:1} = \{\mathbf{x}_t : t \in [0, 1]\}$ to denote a trajectory.

The reverse-time SDE (2) defines a continuous-time Markov chain (CTMC) whose transition law is fully determined by the score $\nabla_\mathbf{x} \log p_{\text{ref}}(\mathbf{x}, t)$ together with $\mathbf{f}$ and $\sigma$.

### 2.2 FORMULATION FROM STOCHASTIC OPTIMAL CONTROL

We adopt the SOC objective with zero running cost and terminal cost $-\eta r(\mathbf{x})$ (terminal reward $r : \mathbb{R}^d \to \mathbb{R}$ with coefficient $\eta > 0$), and define the *control* as the score gap

$$\frac{1}{\sigma(t)}\mathbf{u}(\mathbf{x}, t) := \mathbf{s}_\theta(\mathbf{x}, t) - \nabla_\mathbf{x} \log p_{\text{ref}}(\mathbf{x}, t),$$

so $\mathbf{u} : \mathbb{R}^d \times [0,1] \to \mathbb{R}^d$ has the same dimension as $\mathbf{x}$.

According to the definition in (Domingo-Enrich et al., 2025; Bellman, 1966), the controlled reverse-time SDE is

$$d\mathbf{x}_t^{\mathbf{u}} = \left( \mathbf{b}_0(\mathbf{x}_t^{\mathbf{u}}, t) + \sigma(t)\, \mathbf{u}(\mathbf{x}_t^{\mathbf{u}}, t) \right) dt + \sigma(t)\, d\mathbf{B}_t, \tag{3}$$

where $\mathbf{b}_0(\mathbf{x}, t) := \mathbf{f}(\mathbf{x}, t) + \sigma(t)^2 \nabla_{\mathbf{x}} \log p_{\text{ref}}(\mathbf{x}, t)$, and $\mathbf{x}_t^{\mathbf{u}}$ is the controlled process under $\mathbf{u}$. Let $p^{\mathbf{u}}(\boldsymbol{x} \mid \mathbf{x}_0)$ denote the path measure of the trajectory $\boldsymbol{x}_{0:1}$ induced by (3) given the initial state $\mathbf{x}_0$.

The optimal control problem is to find $\mathbf{u}^*$ minimizing the expected cost:

$$\min_{\mathbf{u}} \ \mathbb{E}_{\boldsymbol{x}^{\mathbf{u}} \sim p^{\mathbf{u}}} \left[ \int_0^1 \tfrac{1}{2} \|\mathbf{u}(\mathbf{x}_t^{\mathbf{u}}, t)\|^2 \, dt - \eta r(\mathbf{x}_1^{\mathbf{u}}) \right],$$

where $\mathbf{x}_1^{\mathbf{u}}$ is the terminal state of the controlled trajectory. With the Theorem 2 in (Domingo-Enrich et al., 2025), we have

$$D_{\text{KL}}\left( p^{\mathbf{u}}(\boldsymbol{x} \mid \mathbf{x}_0) \,\big\|\, p_{\text{ref}}(\boldsymbol{x} \mid \mathbf{x}_0) \right) = \mathbb{E}_{\boldsymbol{x}^{\mathbf{u}} \sim p^{\mathbf{u}}} \left[ \int_0^1 \tfrac{1}{2} \|\mathbf{u}(\mathbf{x}_t^{\mathbf{u}}, t)\|^2 dt \right], \tag{4}$$

which shows that the optimal control problem is equivalent to the KL-regularized continuous-time RL(Doya, 2000) problem:

$$\max_{\mathbf{u} \in \mathcal{U}} \mathbb{E}_{\mathbf{x}_0 \sim p_0} \left[ \mathbb{E}_{\boldsymbol{x} \sim p^{\mathbf{u}}(\cdot \mid \mathbf{x}_0)} \eta r(\mathbf{x}_1^{\mathbf{u}}) - D_{\text{KL}}(p^{\mathbf{u}}(\boldsymbol{x} \mid \mathbf{x}_0) \| p_{\text{ref}}(\boldsymbol{x} \mid \mathbf{x}_0)) \right], \tag{5}$$

where $p_0$ is the prior over $\mathbf{x}_0$ (e.g., $\mathcal{N}(\mathbf{0}, \mathbf{I})$).

## 2.3 DPO FORMULATION OF DIFFUSION MODELS: A BANDIT VIEW

In the bandit view, we regard the diffusion model as a policy that directly produces the final output $\mathbf{x}_1$ with terminal marginal

$$\pi_\theta(\mathbf{x}_1),$$

i.e., $\pi_\theta$ denotes the distribution of the terminal state under the learned policy (not necessarily equal to $p_{\text{ref}}$).

With this interpretation, the DPO(Rafailov et al., 2023) loss is written directly in terms of the final-sample probabilities:

$$\mathcal{L}_{\text{DPO}}(\pi_\theta; \pi_{\text{ref}}) = -\mathbb{E}_{(\mathbf{x}_1^+, \mathbf{x}_1^-) \sim \mathcal{D}} \left[ \log \sigma \left( \frac{1}{\eta} \log \frac{\pi_\theta(\mathbf{x}_1^+)}{\pi_{\text{ref}}(\mathbf{x}_1^+)} - \frac{1}{\eta} \log \frac{\pi_\theta(\mathbf{x}_1^-)}{\pi_{\text{ref}}(\mathbf{x}_1^-)} \right) \right], \tag{6}$$

where $\pi_{\text{ref}}$ is the terminal marginal of the reference process.

**A widely held (but over-simplified) assumption.** We let the discrete time grid be $\mathcal{T}_T := \{0, \frac{1}{T}, \dots, 1\}$, and adopt the reverse-time generative process that starts from $\mathbf{x}_0 \sim \mathcal{N}(\mathbf{0}, \mathbf{I})$ and evolves to the terminal time $\mathbf{x}_1$. It is common in practice to *identify* the marginal likelihood of the final sample with the likelihood of a *single realized trajectory*, i.e.,

$$\pi_\theta(\mathbf{x}_1) \overset{\text{(over-simplified)}}{\approx} p_{\text{ref}}(\mathbf{x}_0) \prod_{t=1}^{T} p_\theta(\mathbf{x}_{\frac{t}{T}} \mid \mathbf{x}_{\frac{t-1}{T}}), \qquad (\star)$$

where the right-hand side is evaluated along one generated path $\mathbf{x}_0 \to \cdots \to \mathbf{x}_1$. However, the right-hand side of $(\star)$ is the *joint* density of a full trajectory $\boldsymbol{x}_{0:1}$, not the *marginal* density of $\mathbf{x}_1$. This notation conflates a path-wise likelihood with the terminal marginal, and underlies many heuristic DPO adaptations.

**The simplification in $(\star)$ obscures two practical issues:**

- **Intractability of the marginal.** The true $\pi_\theta(\mathbf{x}_1)$ requires marginalizing over *all* trajectories that lead to $\mathbf{x}_1$:

$$\pi_\theta(\mathbf{x}_1) = \int \cdots \int p_{\text{ref}}(\mathbf{x}_0) \prod_{t=1}^{T} p_\theta(\mathbf{x}_{\frac{t}{T}} \mid \mathbf{x}_{\frac{t-1}{T}}) \, d\mathbf{x}_{\frac{1}{T} : \frac{T-1}{T}}.$$

  Replacing this integral by a single path likelihood introduces bias.

- **Memory/compute burden.** The horizon $T$ is large(Yang et al., 2023; Liang et al., 2024); storing all intermediate states and their gradients to approximate the marginal (or to backpropagate through all steps) is often impractical in GPU memory.

## 3 PROBLEMS IN CURRENT DIFFUSION DPO METHODS

Therefore, there are mainly three problems in current diffusion DPO methods:

- **Inconsistency between trajectory-level probability and final state probability.** Current methods (Wallace et al., 2023; Liang et al., 2024; Yang et al., 2023) assume that the trajectory-level probability can be approximated by the final state probability, which is not always true. This assumption can lead to suboptimal policies that do not align well with the true reward function.
- **Inefficiency in utilizing reward signals.** Current methods do not fully leverage the reward signals available at each time step, or not pay attention to the consistency of rewards at different time steps, leading to inefficient learning and suboptimal performance.
- **Lack of theoretical guarantees.** Many existing approaches (Yang et al., 2023; Liang et al., 2024) lack rigorous theoretical analysis, making it difficult to understand their convergence properties and optimality. This can lead to uncertainty about the reliability and robustness of the learned policies, especially compared with original DPO (Rafailov et al., 2023) in NLP.

## 4 METHODOLOGY

To address the limitations of existing diffusion DPO methods, we introduce a two-stage framework that leverages the optimal value function as the return for short trajectory segments. This decomposition enables long trajectories to be broken into manageable pieces, where preferences can be assessed more reliably.

### 4.1 OVERALL PIPELINE

Our pipeline consists of two stages. In the first stage, we train the optimal value-distribution function. In the second stage, we perform Value-as-Return Direct Preference Optimization (VRPO) using this learned value function.

The framework is built on two key insights. First, **when trajectories are sufficiently short, the preference over the final state coincides with the preference over the entire trajectory, conditioned on the same starting point.** Moreover, the memory burden is also alleviated. This observation justifies segmenting long trajectories into shorter ones, on which DPO can be applied directly. Second, **the optimal value function admits a closed-form expression that depends only on the reference policy and the terminal reward in the image domain.** As a result, we can learn a value-distribution function to estimate segment-level preferences, and then employ these preferences to guide VRPO training.

### 4.2 FIRST STAGE: TRAINING THE OPTIMAL VALUE-DISTRIBUTION FUNCTION

**Value-distribution definition (terminal at $t = 1$).** We define the *optimal value-distribution function* through the random variable

$$Z(\mathbf{x}_t, t) := r(\mathbf{x}_1), \qquad \mathbf{x}_1 \sim p_{\text{ref}}(\mathbf{x}_1 \mid \mathbf{x}_t, t), \quad t \in \mathcal{T}_T,$$

where $p_{\text{ref}}(\mathbf{x}_1 \mid \mathbf{x}_t, t)$ is the conditional distribution of the reference diffusion model from intermediate state $(\mathbf{x}_t, t)$ to the terminal state $(\mathbf{x}_1, 1)$, and $r(\cdot)$ is the image-domain reward defined at $t = 1$. Then, we directly have the following closed-form expression for the optimal value function:

**Lemma 1.** *The optimal value function of Equation* (5) *admits the closed-form*

$$V^*(\mathbf{x}_t, t, \eta) = \log \mathbb{E}\big[\exp\big(\eta\, Z(\mathbf{x}_t, t)\big)\big] = \log \mathbb{E}_{\mathbf{x}_1 \sim p_{\text{ref}}(\mathbf{x}_1 \mid \mathbf{x}_t, t)} \big[\exp\big(\eta\, r(\mathbf{x}_1)\big)\big],$$

*which depends only on the reference policy $p_{\text{ref}}$ and the terminal reward $r$. Consequently, once the distribution $Z(\mathbf{x}_t, t)$ is learned, the KL coefficient $\frac{1}{\eta}$ can be adjusted* post hoc *without retraining.*

We explored multiple approaches for modeling the distribution of $Z(\mathbf{x}_t, t)$, and observed that training with a mix of the Continuous Ranked Probability Score (CRPS) loss, Binary Cross Entropy loss and smooth loss to fit the cumulative distribution function (CDF) yields more stable performance than directly fitting the probability density function (PDF) using maximum likelihood estimation (MLE) or KL divergence, as shown in Algorithm 2. Accordingly, we parameterize a model

$F_\phi(z \mid \mathbf{x}_t, t)$ to approximate the CDF of $Z(\mathbf{x}_t, t)$, and compute the optimal value function for different $\eta$ as

$$V_\phi^*(\mathbf{x}_t, t, \eta) \;=\; \log \int e^{\eta z} \, dF_\phi(z \mid \mathbf{x}_t, t).$$

### 4.3 SECOND STAGE: STAGE-WISE VRPO TRAINING

We decompose the long trajectory on $\mathcal{T}_T$ into short segments of length $L$ steps, so that $t_k \in \{0, \frac{1}{T}, \ldots, 1 - \frac{L}{T}\}$ and the segment runs from $t_k$ to $t_k + \frac{L}{T}$. Under the short-segment assumption, *the terminal-state preference within the segment aligns with the trajectory preference given the same start*. We therefore evaluate segment preferences using the optimal value function at the segment endpoint. The overall procedure is shown in Algorithm 1.

---

**Algorithm 1** Stage-wise VRPO with Learned Value $V_\phi^*$ (time grid $0, \frac{1}{T}, \ldots, 1$)

---

**Require:** Reference transition $p_{\text{ref}}(\mathbf{x}_{t+\frac{1}{T}} \mid \mathbf{x}_t)$; learned $q_\phi(z \mid \mathbf{x}_t, t)$; model $p_\theta$ to train; segment
    length $L$; #segments $K$; DPO temperature $\beta = \frac{1}{\eta}$

1: **Value from the learned distribution:**

$$V_\phi^*(\mathbf{x}_t, t, 1/\beta) \;=\; \log \mathbb{E}_{z \sim q_\phi(\cdot \mid \mathbf{x}_t, t)}\big[ \exp\big((1/\beta)\, z\big)\big].$$

2: **for** each training iteration **do**                 ▷ Stage-wise over short segments
3:     Sample start time $t_k \in \{0, \frac{1}{T}, \ldots, 1 - \frac{L}{T}\}$ and start state $\mathbf{x}_{t_k} \sim p_{\text{ref}}(\mathbf{x}_{t_k}, t_k)$
4:     From $\mathbf{x}_{t_k}$, **roll out** $L$ **steps** with the *reference* model to the endpoint $\mathbf{x}_{t_k + \frac{L}{T}}$
5:     Generate candidate endpoints (e.g., two independent rollouts) $\mathbf{x}_{t_k + \frac{L}{T}}^+$ and $\mathbf{x}_{t_k + \frac{L}{T}}^-$ and **rank**
    by

$$V_\phi^*(\mathbf{x}_{t_k + \frac{L}{T}}^+, t_k + \tfrac{L}{T}, \tfrac{1}{\beta}) \;>\; V_\phi^*(\mathbf{x}_{t_k + \frac{L}{T}}^-, t_k + \tfrac{L}{T}, \tfrac{1}{\beta}).$$

6:     Update $p_\theta$ with the **VRPO** loss:

$$\mathcal{L}_{\text{VRPO}} \;=\; -\mathbb{E}\left[\log \sigma\left( \beta \log \frac{p_\theta\left(\mathbf{x}_{t_k + \frac{L}{T}}^+ \mid \mathbf{x}_{t_k}\right)}{p_{\text{ref}}\left(\mathbf{x}_{t_k + \frac{L}{T}}^+ \mid \mathbf{x}_{t_k}\right)} - \beta \log \frac{p_\theta\left(\mathbf{x}_{t_k + \frac{L}{T}}^- \mid \mathbf{x}_{t_k}\right)}{p_{\text{ref}}\left(\mathbf{x}_{t_k + \frac{L}{T}}^- \mid \mathbf{x}_{t_k}\right)} \right)\right].$$

7: **end for**
8: **Output:** A VRPO–trained diffusion model $p_\theta$.

---

**Lemma 2.** *Suppose $\frac{L}{T} = \Delta$ is sufficiently small, then we have that given the same start $(\mathbf{x}_{t_k}, t_k)$, the endpoint probability can be approximated by a one-step (Euler–Maruyama) approximation:*

$$p(\mathbf{x}_{t_k + \frac{L}{T}} \mid \mathbf{x}_{t_k}, t_k) \approx \mathcal{N}\big(\mu_{t_k}, \, \Sigma_{t_k}\big),$$

*where*

$$\mu_{t_k} = \mathbf{x}_{t_k} + \big(f(\mathbf{x}_{t_k}, t_k) + \sigma(t_k)^2 s_\theta(\mathbf{x}_{t_k}, t_k)\big)\Delta, \quad \Sigma_{t_k} = \sigma(t_k)^2 \, \Delta \, \mathbf{I}.$$

We use Lemma 2 to ensure that the endpoint distribution of a short trajectory segment can be well approximated. In practice, however, one may also work with the trajectory probability because of the short trajectory. Then, we have:

**Theorem 3** (Optimality of VRPO). *Consider a trajectory on $\mathcal{T}_T$ that is decomposed into $K$ short segments, each of length $L$ with $\Delta = \frac{L}{T}$ sufficiently small. If VRPO is applied to segment-level preferences defined by $V_k^*(\mathbf{x}_t, t, \eta)$, then under sufficient model capacity, the resulting model is equivalent to training directly on the optimal target distribution*

$$p(\mathbf{x}_1) \;\propto\; p_{data}(\mathbf{x}_1) \, \exp\big(\eta\, r(\mathbf{x}_1)\big).$$

Table 1: Results of quantitative comparison with baselines and ablation studies on SDXL.

| Methods | PickScore | ImageReward | Aesthetic Score | HPSv2 |
|---|---|---|---|---|
| SDXL | 21.7093 | 0.5887 | 6.4658 | 0.2738 |
| MAPO | 21.9038 | 0.7429 | 6.5918 | 0.2791 |
| DiffusionDPO | 22.3269 | 0.9411 | 6.3959 | 0.2837 |
| SPO | 22.7821 | 1.0536 | 6.8035 | 0.2858 |
| VRPO (Ours) | **23.0656** | **1.2404** | **7.1894** | **0.2867** |
| Full Methods | 23.0656 | **1.2404** | **7.1894** | **0.2867** |
| w/o OVF | 22.9077 | 1.1439 | 6.8553 | 0.2825 |
| CE Loss for Logits | 22.6795 | 1.0325 | 6.7490 | 0.2832 |
| Average Scores | 22.9472 | 1.1308 | 6.8737 | 0.2843 |
| Bins Width 20 | 22.4321 | 1.0172 | 6.7177 | 0.2825 |
| Bins Width 30 | 22.8641 | 1.1605 | 6.7899 | 0.2823 |
| Bins Width 75 | 23.0070 | 1.2186 | 7.1854 | 0.2839 |
| Bins Width 100 | **23.0923** | 1.2008 | 7.0816 | 0.2859 |
| w/o OVF + Trajectory | 22.6980 | 1.0378 | 6.7569 | 0.2829 |

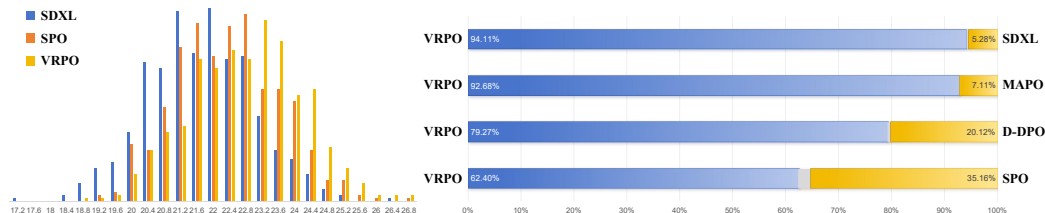

Figure 2: *Left*: Histograms of PickScore values for SDXL, SPO, and VRPO. *Right*: Win rates of VRPO compared to other baselines in terms of PickScore values.

## 5 EXPERIMENTS

### 5.1 IMPLEMENTATION DETAILS

#### 5.1.1 VALUE FUNCTION TRAINING

For the CDF model $F_\phi(z \mid \mathbf{x}_t, t)$, we adopt the PickScore v1 model (Kirstain et al., 2023) as our backbone. PickScore builds on the CLIP (Radford et al., 2021) dual-encoder architecture, where a text encoder and an image encoder map prompts and images into a shared embedding space, and their dot product (scaled by a learnable parameter) serves as the preference score. The model is fine-tuned on human preference data using a pairwise ranking loss, aligning the scores with users' comparative judgments of image quality. In our framework, the parameters of the PickScore encoders are kept frozen, so that the extracted text and image embeddings remain fixed. On top of these embeddings, we design a lightweight prediction head: a timestep embedding is first combined with the image embedding, and the resulting representation is fused with the text embedding through a MLP. Finally, the fused features are passed into a monotonic CDF (cumulative distribution function) Head, which enforces non-decreasing outputs along discretized thresholds and produces a valid CDF output.

We train the model following Algorithm 2. Instead of rolling out a trajectory at each training step, we construct a dataset of images $\mathbf{x}_1$ using the prompts from the training split of the Pic-a-Pic v1 dataset (Kirstain et al., 2023). During training, these images are further perturbed with DDIM (Song et al., 2021a) to obtain $\mathbf{x}_t$, which are then used as inputs. The PickScore value serves as the ground-truth reward $r(\mathbf{x}_1)$. By default, we set the horizon to $T = 20$, and the number of discrete classes to $M = 50$. For the SD-1.5 (Rombach et al., 2022) experiments, PickScore values are uniformly discretized into 50 bins over the range [15, 25], while for SDXL (Podell et al., 2023) the range [17,

Table 2: Results of quantitative comparison with baselines and ablation studies on SD-1.5.

| Methods | PickScore | ImageReward | Aesthetic Score | HPSv2 |
|---|---|---|---|---|
| SD v1.5 | 20.346 | -0.089 | 5.956 | 0.235 |
| SPO | 21.1982 | 0.2726 | 6.4497 | 0.2712 |
| Ours | **21.7609** | **0.6704** | **6.5209** | **0.2803** |
| w/o OVF | 21.7183 | 0.5544 | 6.4832 | 0.2770 |
| w/o OVF + Trajectory | 21.5579 | 0.4285 | 6.4681 | 0.2733 |

27] is divided into 50 bins. Training is performed with the AdamW (Loshchilov & Hutter, 2017) optimizer, using a learning rate of $1 \times 10^{-4}$, betas $(0.9, 0.98)$, $\epsilon = 1 \times 10^{-6}$, and a weight decay of $0.01$. We adopt a learning rate schedule with warm-up followed by cosine annealing. The overall loss is defined as a weighted sum of three components: a CRPS loss with weight $0.7$, a cross-entropy loss with weight $0.3$, and a first-order total variation smoothness loss.

### 5.1.2 VRPO TRAINING

We train the VRPO on the training prompts of the Pick-a-Pic dataset. We employ DDIM sampling with 20 denoising steps, combined with classifier-free guidance at a scale of 5.0. Both SD-1.5 and SDXL are fine-tuned using LoRA (Hu et al., 2021) adapters for 15 epochs, with the LoRA rank set to 4 for SD-1.5 and 64 for SDXL. A regularization coefficient of $\beta = 10$ (with $\eta = \frac{1}{\beta} = 0.1$) is applied throughout training. For SD-1.5, we adopt a batch size of 40 and a learning rate of $2 \times 10^{-5}$, while for SDXL we use a batch size of 8 with gradient accumulation over 2 steps and a learning rate of $1.5 \times 10^{-5}$. Optimization is performed with AdamW, configured with $\beta_1 = 0.9$, $\beta_2 = 0.999$, and a weight decay of $1 \times 10^{-4}$. The number of inner optimization steps $L$ is set to 1 for SD-1.5 and 4 for SDXL.

### 5.2 COMPARING WITH BASELINES

#### 5.2.1 QUANTITATIVE COMPARISON

**Problem Settings.** We compare our method VRPO against three strong baselines: MAPO (She et al., 2024), Diffusion-DPO (Wallace et al., 2023), and SPO (Liang et al., 2024). MAPO and Diffusion-DPO are trained on the Pick-a-Pic v2 dataset, which contains over 800,000 image–preference pairs, whereas SPO and our method use the Pick-a-Pic v1 dataset with about 500,000 pairs. For fairness, we adopt the official checkpoints of all baselines. Our evaluations are performed on the "validation_unique" split of Pick-a-Pic v1: after filtering out prompts with special characters, we obtain 492 valid prompts.

We adopt four evaluation metrics: PickScore v1, ImageReward (Xu et al., 2023), Aesthetic Score (Schuhmann, 2022), and HPSv2 (Wu et al., 2023). PickScore v1 is a CLIP-based preference predictor trained on human data, and has been shown to correlate more strongly with human rankings than earlier automated metrics. HPSv2 is a preference prediction model trained on the large-scale HPD v2 dataset, designed to better capture human judgments across diverse image distributions. Aesthetic Score evaluates the visual appeal of generated images in a prompt-agnostic manner, while ImageReward aggregates multiple quality features to provide a holistic assessment of image quality.

**Quantitative comparison.** The results of the quantitative comparison are reported in Table 1 and Table 2. To provide a more intuitive understanding of the superiority of VRPO, we further present the histogram of PickScore values and the win rate comparison in Figure 2. When applied to SD v1.5, our method outperforms all other methods across all metrics by a substantial margin. For SDXL, our method demonstrates clear advantages in PickScore, ImageReward, and Aesthetic Score. Notably, in the supervised metric of PickScore, we outperform the baseline by more than 0.65 on SD v1.5 and 0.19 on SDXL. When comparing with the best-performing baseline, the SPO method, our approach uses significantly fewer parameters for training the value function but achieves superior results. This further supports the our effectiveness of utilizing reward signals. Specifically, SPO incorporates the timestep into the input of the CLIP image encoder and fine-tunes the entire CLIP

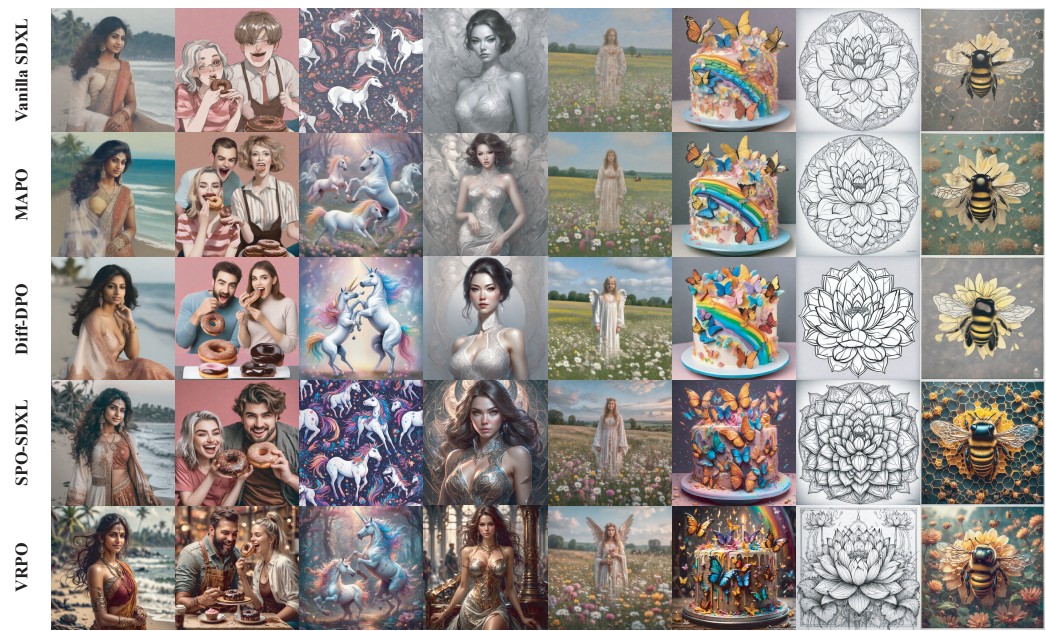

Figure 3: Comparison with baseline methods. See Appendix I for the corresponding prompts.

model. In contrast, we freeze the CLIP encoder, and finetune a light-weight head. Remarkably, while delivering state-of-the-art performance, VRPO also consumes the least training time across all baselines (see detailed comparison in Appendix G).

### 5.2.2 QUALITATIVE COMPARISON.

Figure 3 presents qualitative comparisons across different methods. All images are generated with the same random seed to enable a direct visual comparison of the effects of different fine-tuning strategies. Compared with the baselines, VRPO produces images with richer details and more coherent compositions.

Table 3: Preference comparison of VRPO against baseline methods (SDXL and SPO) in terms of overall preference, visual quality, and text–image alignment (reported as Win / Draw / Lose).

| Methods | Overall Performance | Visual Quality | Text Alignment |
|---|---|---|---|
| VRPO v.s SDXL | **80.6**% / 0% / 19.4% | **76.8**% / 4.1% / 19.1% | **55.4**% / 17.6 % / 27.0% |
| VRPO v.s SPO | **65.8**% / 0% / 34.2% | **50.2**% / 26.4% / 23.4% | **40.0**% / 22.5% / 37.5% |

### 5.2.3 HUMAN PREFERENCE COMPARISON USING MLLM.

In this study, we employed ChatGPT-5.0 as an evaluator to assess the quality of generated images by interpreting and articulating human-like judgments. Following the experimental setup of SPO, we randomly sampled 100 prompts from Parti-Prompts (Yu et al., 2022) and 200 prompts from HPSv2 (Wu et al., 2023), both unseen by VRPO and SPO, covering a broad range of categories. GPT-5.0 was then instructed to compare the outputs of SPO and our method along three dimensions: overall preference, visual appeal, and text–image alignment. More details are provided in Appendix J.

The results shown in Table 3 indicate that our method consistently received more "winning votes" from users in both general preference and visual appeal. Notably, the margin by which our method outperforms SPO in terms of visual appeal is particularly substantial. Please note that we also provided objective evaluation metrics in Table 4 in Appendix. These metrics revealed that our method demonstrated even more pronounced advantages compared to the results obtained from the Pick-a-Pic dataset.

## 5.3 ABLATION STUDY

We conduct a series of ablation experiments to validate the effectiveness of our two key design choices—(i) training the value function with rollout trajectories rather than unrelated image datasets (as in SPO), and (ii) adapting the optimal value function (OVF). Results are reported on both SD-1.5 and SDXL (see Table 2 and Table 1), with more detailed ablations provided on SDXL.

**Impact of probabilistic distribution.** We remove the optimal value function (*w/o OVF*) completely and train a regression-based value function directly on the rollout trajectory dataset, we observe consistent performance degradation across all evaluation metrics on both SD-1.5 and SDXL. This confirms that modeling the inherent uncertainty in diffusion sampling through a probabilistic value function is crucial for improved performance.

**Loss formulation.** To further examine the effect of the training objective, we replace the CRPS loss with only the cross-entropy loss and substitute the CDF Head with a simple probability head (*CE Loss for Logits*). This modification results in a significant performance drop, indicating that cross-entropy loss alone fails to provide stable convergence for the value function.

**Effectiveness of optimal value function.** Instead of using the optimal score function derived in Lemma 1, we also experiment with computing the average score of the predicted distribution (*Average Scores*). This approximation again results in degraded performance, demonstrating that the optimal score function enhances the ability of DPO methods to effectively leverage the reward.

**Effect of bin width.** We study the discretization granularity of the distribution by varying the number of bins (*Bin Width*). Given that PickScore values span a range of 10 for both SD-1.5 ([15, 25]) and SDXL ([17, 27]), reducing the number of bins to 20 or 30 causes a severe performance decline, whereas increasing the bin resolution beyond 50 yields no further noticeable improvements. This suggests that around 50 bins strike a good balance between expressiveness and efficiency.

**Joint removal of contributions.** Finally, we remove both the roll-out-trajectory dataset and the probabilistic formulation (*w/o OVF + Trajectory*), which leads to a drastic performance drop. Comparing this result with *w/o OVF*, we observe that the dataset alone also contributes substantially to the overall performance.

More ablation studies on segment length and miss-ranking rate are provided on Appendix H.

## 6 DISCUSSION

The proposed value-distribution function is not tied to a specific optimization scheme. It can be directly employed within GRPO(Shao et al., 2024) or other preference-based objectives, and it naturally extends to classical policy-gradient methods such as REINFORCE(Williams, 1992), where it acts as a baseline to reduce variance. This flexibility suggests that our framework is broadly compatible with a wide family of reinforcement learning algorithms.

A further distinction arises when comparing reinforcement learning in diffusion models to reinforcement learning in natural language processing. In NLP, policies are discrete and autoregressive, which makes trajectory probabilities tractable. Diffusion models, by contrast, evolve under continuous-time stochastic dynamics, where the marginal distribution of the final state requires integrating over all possible paths. Our framework addresses this gap by reformulating the problem in terms of value functions, thereby avoiding intractable marginalization while maintaining theoretical alignment with the target tilted distribution.

## 7 CONCLUSION

We presented a two-stage reinforcement learning framework for diffusion models that leverages the optimal value function as the return signal for short trajectory segments. By first learning a value-distribution function and then applying stage-wise VRPO, our method provides a principled approach to align diffusion models with image-domain rewards. We established theoretical guarantees showing that, under sufficient model capacity, the resulting model is equivalent to training on the tilted distribution proportional to $p(\mathbf{x}) \exp(\eta r(\mathbf{x}))$. Empirical results on large-scale diffusion models confirm the effectiveness of our approach, yielding stable and consistent improvements over prior DPO-based methods.

## 8 REPRODUCIBILITY STATEMENT

We provide evaluation scripts and pre-trained checkpoints at the following *anonymous* webpage: https://osf.io/7a4fh/?view_only=f951fa68c7ef43a19a44ee38b11847f6. The page includes instructions for environment setup, exact command lines for evaluation, and expected outputs for sanity checks. Once the paper is made public, we will also open-source the full code repository.

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

## A    ETHICS STATEMENT

This paper studies methods for preference optimization in diffusion models. Our contributions are mainly methodological. Potential positive impacts include improving controllability of generative models, which may benefit applications such as art creation and scientific visualization. However, misuse of the method could lead to harmful content generation or reinforce biases present in the training data. We rely only on publicly available datasets in our experiments, and we encourage future work to carefully consider fairness, safety, and data governance issues when deploying such models.

## B    USE OF LLMS

We used large language models (LLMs) solely for grammar and spelling checking in the preparation of this paper. No LLMs were involved in generating research ideas, conducting experiments, analyzing results, or writing technical content. All scientific contributions and claims are entirely the work of the authors.

## C    RELATED WORK

**Diffusion models.**    Diffusion probabilistic models (Ho et al., 2020; Song et al., 2021b;a) have emerged as state-of-the-art generative models across image, audio, and video domains. Their success largely stems from stable likelihood training and scalable architectures, with large-scale variants such as Stable Diffusion (Rombach et al., 2022; Podell et al., 2023) demonstrating impressive generative capabilities. Recent advances (Ho & Salimans, 2022; Meng et al., 2021; Zhang et al., 2023; Nichol et al., 2021; Lee et al., 2025; Liu et al., 2024; Sun et al., 2024) have sought to improve controllability by conditioning on textual prompts, sketches, or layout, but aligning with more complex human-defined reward signals remains a challenge.

**Reinforcement learning for generative models.**    RL has been widely adopted in language modeling (Rafailov et al., 2023; Ouyang et al., 2022), where it enables alignment with human preferences through pairwise comparisons. In diffusion models, early efforts adapted policy gradient (Agarwal et al., 2019; Black et al., 2023), but these methods often suffer from high variance and unstable credit assignment across long horizons. Recent works therefore explore preference-based training tailored to diffusion models, such as Diffusion-DPO (Wallace et al., 2023) and D3PO (Yang et al., 2023), which directly optimize model likelihood ratios against a reference. However, these approaches typically equate trajectory likelihoods with terminal-state probabilities, an oversimplification that can misalign the learned distribution with the intended reward structure.

**Segment-wise training.**    To mitigate horizon-related difficulties, several works decompose long diffusion rollouts into shorter segments (Liang et al., 2024; Yang et al., 2023). While this improves stability, segment-level objectives are often designed heuristically, and their consistency with the global reward remains unclear. Our work provides a principled alternative: we ground the segment-wise objective in stochastic optimal control, derive a closed-form expression of the optimal value function, and use it to define preference-consistent objectives for Direct Preference Optimization. This yields both theoretical guarantees and practical improvements.

**Stochastic optimal control and continuous-time RL.**    Our approach is also connected to the literature on SOC and continuous-time RL. Linearly-solvable MDPs (Todorov, 2006; 2009) show that optimal controls can often be expressed via exponentiated value functions and Doob $h$-transforms, a perspective that closely parallels our formulation. In continuous-time settings, stochastic control theory provides tools such as the Hamilton–Jacobi–Bellman equation and Girsanov's theorem (Fleming & Soner, 2006; Oksendal, 2013), which describe how optimal policies reshape diffusion dynamics. Recent works (Domingo-Enrich et al., 2025) have further connected RL objectives with score-based generative modeling. Our method builds on these insights by importing SOC principles into diffusion preference optimization, establishing both theoretical grounding and practical benefits.

# D  THE TRANING ALGORITHM OF VALUE-DISTRIBUTION FUNCTION

We have a pratical training algorithm as shown in Algorithm 2.

---

**Algorithm 2** Learning the Value-Distribution via CDF Modeling (CRPS + BCE + TV-Smooth)

---

**Require:** Reference sampler $p_{\text{ref}}(\mathbf{x}_1)$; reward $r(\mathbf{x}_1)$; forward noising $\text{Noisify}(\mathbf{x}_1, t)$; CDF model
  $F_\phi(z \mid \mathbf{x}_t, t)$; grid $\{c_m\}_{m=1}^M$; weights $(\lambda_{\text{crps}}, \lambda_{\text{bce}}, \lambda_{\text{tv}})$; small $\varepsilon > 0$
 1: **for** each iteration **do**
 2:   **Sample terminal images:** $\{\mathbf{x}_1^{(i)}\}_{i=1}^B \sim p_{\text{ref}}(\mathbf{x}_1)$
 3:   **Score images:** $y^{(i)} \leftarrow r(\mathbf{x}_1^{(i)})$
 4:   **Sample time and noising:** $t^{(i)} \sim \text{Unif}(\mathcal{T}_T)$, $\mathbf{x}_{t^{(i)}}^{(i)} \leftarrow \text{Noisify}(\mathbf{x}_1^{(i)}, t^{(i)})$
 5:   **Evaluate CDF on grid:** $F_m^{(i)} \leftarrow F_\phi(c_m \mid \mathbf{x}_{t^{(i)}}^{(i)}, t^{(i)})$, $m = 1{:}M$
 6:   **CRPS loss:**
$$\mathcal{L}_{\text{crps}} \;=\; \frac{1}{B} \sum_{i=1}^B \sum_{m=1}^M \left( F_m^{(i)} - \mathbf{1}\{c_m \geq y^{(i)}\} \right)^2$$
 7:   **CDF binary cross-entropy:**
$$\mathcal{L}_{\text{bce}} \;=\; -\frac{1}{B} \sum_{i=1}^B \sum_{m=1}^M \left[ \mathbf{1}\{c_m \geq y^{(i)}\} \log(F_m^{(i)} + \varepsilon) + \left(1 - \mathbf{1}\{c_m \geq y^{(i)}\}\right) \log(1 - F_m^{(i)} + \varepsilon) \right]$$
 8:   **TV smoothness on CDF:**
$$\mathcal{L}_{\text{tv}} \;=\; \frac{1}{B} \sum_{i=1}^B \frac{1}{M-1} \sum_{m=1}^{M-1} \left| F_{m+1}^{(i)} - F_m^{(i)} \right|$$
 9:   **Total loss and update:**
$$\mathcal{L} \;=\; \lambda_{\text{crps}} \mathcal{L}_{\text{crps}} + \lambda_{\text{bce}} \mathcal{L}_{\text{bce}} + \lambda_{\text{tv}} \mathcal{L}_{\text{tv}}, \qquad \phi \leftarrow \phi - \eta \nabla_\phi \mathcal{L}$$
10: **end for**
11: **Output:** Trained CDF $F_\phi(z \mid \mathbf{x}_t, t)$.
12: **Post hoc value:**
$$V_\phi^*(\mathbf{x}_t, t, \eta) \;\approx\; \log \sum_{m=1}^M \tilde{\pi}_m(\mathbf{x}_t, t)\, e^{\eta c_m},$$

$$\tilde{\pi}_m(\mathbf{x}_t, t) = \frac{\left( F_\phi(c_m \mid \mathbf{x}_t, t) - F_\phi(c_{m-1} \mid \mathbf{x}_t, t) \right)_+ + \varepsilon}{\sum_{\ell=1}^M \left( F_\phi(c_\ell \mid \mathbf{x}_t, t) - F_\phi(c_{\ell-1} \mid \mathbf{x}_t, t) \right)_+ + M\varepsilon}.$$

---

# E  KEY ANALYSIS RESULTS FROM SOC

## E.1  OPTIMAL VALUE FUNCTION

From the SOC formulation, the *value function* is defined as the optimal objective under the controlled reverse-time SDE (3):

$$V(\mathbf{x}, t) \;:=\; \min_{\mathbf{u}} \; \mathbb{E}_{\boldsymbol{x}^{\mathbf{u}} \sim p^{\mathbf{u}}} \left[ \int_t^1 \tfrac{1}{2} \|\mathbf{u}(\mathbf{x}_s^{\mathbf{u}}, s)\|^2 \, \mathrm{d}s - \eta r(\mathbf{x}_1^{\mathbf{u}}) \,\big|\, \mathbf{x}_t^{\mathbf{u}} = \mathbf{x} \right]. \tag{7}$$

Equivalently, by the linearly-solvable control result (Kappen, 2005), the value function admits a representation with respect to the uncontrolled reference process $p_{\text{ref}}$:

$$V(\mathbf{x}, t) \;=\; -\log \mathbb{E}_{\boldsymbol{x} \sim p_{\text{ref}}} \left[ \exp\!\big(\eta r(\mathbf{x}_1)\big) \,\big|\, \mathbf{x}_t = \mathbf{x} \right], \tag{8}$$

where the expectation is over trajectories $\boldsymbol{x}_{0:1}$ of the base process.

### E.2 OPTIMAL SCORE FUNCTION

The optimal control $\mathbf{u}^*(\mathbf{x}, t)$ can be expressed via the gradient of the value function (Kappen, 2005):

$$\mathbf{u}^*(\mathbf{x}, t) = -\sigma(t) \nabla_{\mathbf{x}} V(\mathbf{x}, t). \tag{9}$$

Then the *optimal score function* is

$$\mathbf{s}^*(\mathbf{x}, t) = \nabla_{\mathbf{x}} \log p_{\text{ref}}(\mathbf{x}, t) - \nabla_{\mathbf{x}} V(\mathbf{x}, t). \tag{10}$$

It is worth noting that $\mathbf{s}^*(\mathbf{x}, t)$ corresponds to the score of the tilted distribution

$$p^*(\mathbf{x}) = \frac{1}{Z} p_{\text{ref}}(\mathbf{x}) \exp(\eta r(\mathbf{x})),$$

where $Z$ is the normalization constant.

## F PROOFS

### F.1 PROOF OF LEMMA 1

*Proof.* Fix $(\mathbf{x}_t, t)$ and let $\mathbb{P}^{\text{ref}}_{\mathbf{x}_t, t}$ denote the (path) law on trajectories $(\mathbf{x}_s)_{s \in [t,1]}$ induced by the (reference) reverse-time SDE starting at $\mathbf{x}_t$ at time $t$. For any admissible control $\mathbf{u}$, let $\mathbb{P}^{\mathbf{u}}_{\mathbf{x}_t, t}$ be the corresponding controlled path law. By Girsanov's theorem (under the usual linear growth/Lipschitz assumptions ensuring absolute continuity), we have

$$D_{\text{KL}}\Big(\mathbb{P}^{\mathbf{u}}_{\mathbf{x}_t, t} \,\big\|\, \mathbb{P}^{\text{ref}}_{\mathbf{x}_t, t}\Big) = \mathbb{E}_{\mathbb{P}^{\mathbf{u}}_{\mathbf{x}_t, t}} \left[ \int_t^1 \tfrac{1}{2} \|\mathbf{u}(\mathbf{x}_s, s)\|^2 \, ds \right].$$

Hence the KL–regularized continuous-time RL objective (Equation (5)) can be written as

$$V^*(\mathbf{x}_t, t, \eta) = \sup_{\mathbf{u}} \Big\{ \eta \, \mathbb{E}_{\mathbb{P}^{\mathbf{u}}_{\mathbf{x}_t, t}}[r(\mathbf{x}_1)] - D_{\text{KL}}(\mathbb{P}^{\mathbf{u}}_{\mathbf{x}_t, t} \,\|\, \mathbb{P}^{\text{ref}}_{\mathbf{x}_t, t}) \Big\}. \tag{11}$$

According to Equation (8), we have

$$V^*(\mathbf{x}_t, t, \eta) = \mathbb{E}_{\mathbf{x}_1 \sim p_{\text{ref}}(\mathbf{x}_1 | \mathbf{x}_t, t)} \big[ \exp\big(\eta \, r(\mathbf{x}_1)\big) \big],$$

which yields the claimed closed form. The expression depends only on the reward distribution given $\mathbf{x}_t$, so the coefficient $1/\eta$ (KL temperature) can be tuned *post hoc* without retraining. $\square$

### F.2 PROOF OF LEMMA 2

*Proof.* Consider the reverse-time SDE on $[t_k, t_k + \Delta]$:

$$d\mathbf{x}_t = \Big(f(\mathbf{x}_t, t) + \sigma(t)^2 s_\theta(\mathbf{x}_t, t)\Big) dt + \sigma(t) \, d\mathbf{B}_t, \qquad \mathbf{x}_{t_k} \text{ given}.$$

Assume the usual Lipschitz and linear-growth conditions so that the Euler–Maruyama (EM) scheme is valid. Over a small step $\Delta = \frac{L}{T}$, the EM update at $(\mathbf{x}_{t_k}, t_k)$ reads

$$\mathbf{x}_{t_k + \Delta} \approx \mathbf{x}_{t_k} + \Big(f(\mathbf{x}_{t_k}, t_k) + \sigma(t_k)^2 s_\theta(\mathbf{x}_{t_k}, t_k)\Big)\Delta + \sigma(t_k)\big(\mathbf{B}_{t_k + \Delta} - \mathbf{B}_{t_k}\big).$$

Since $\mathbf{B}_{t_k + \Delta} - \mathbf{B}_{t_k} \sim \mathcal{N}(\mathbf{0}, \Delta \, \mathbf{I})$ and is independent of $\mathbf{x}_{t_k}$, conditioning on the same start $(\mathbf{x}_{t_k}, t_k)$ implies

$$\mathbf{x}_{t_k + \Delta} \,\big|\, \mathbf{x}_{t_k} \approx \mathcal{N}\Big( \underbrace{\mathbf{x}_{t_k} + \big(f(\mathbf{x}_{t_k}, t_k) + \sigma(t_k)^2 s_\theta(\mathbf{x}_{t_k}, t_k)\big)\Delta}_{\mu_{t_k}}, \, \underbrace{\sigma(t_k)^2 \Delta \, \mathbf{I}}_{\Sigma_{t_k}}\Big).$$

Standard EM error bounds give $\|\mathbb{E}[\varphi(\mathbf{x}_{t_k + \Delta})] - \mathbb{E}[\varphi(\mathbf{x}^{\text{EM}}_{t_k + \Delta})]\| = \mathcal{O}(\Delta)$ for smooth test functions $\varphi$ (weak order 1), and $\mathbb{E}\|\mathbf{x}_{t_k + \Delta} - \mathbf{x}^{\text{EM}}_{t_k + \Delta}\|^2 = \mathcal{O}(\Delta)$ (strong order 1/2), so the Gaussian approximation holds to first order in $\Delta$. $\square$

### F.3 PROOF OF THEOREM 3

We decompose the proof into three parts: the characterization of the segment-level optimum, the inductive propagation of optimal marginals, and the base case at $t = 0$.

Define the *desirability* (exponentiated value) at any $(\mathbf{x}, t)$ by

$$z^*(\mathbf{x}, t; \eta) := \exp\big(V^*(\mathbf{x}, t, \eta)\big) = \mathbb{E}_{\mathbf{x}_1 \sim p_{\mathrm{ref}}(\cdot \mid \mathbf{x}, t)}\big[\exp\big(\eta\, r(\mathbf{x}_1)\big)\big].$$

By the Markov property of $p_{\mathrm{ref}}$ and the law of total expectation, $z^*$ satisfies the *multiplicative Bellman equation* for any intermediate time $t$ and next time $t+\Delta$:

$$z^*(\mathbf{x}, t; \eta) = \mathbb{E}_{\mathbf{x}' \sim p_{\mathrm{ref}}(\cdot \mid \mathbf{x}, t)}\big[z^*(\mathbf{x}', t+\Delta; \eta)\big], \qquad V^*(\mathbf{x}, t, \eta) = \log \mathbb{E}_{\mathbf{x}' \sim p_{\mathrm{ref}}(\cdot \mid \mathbf{x}, t)}\big[e^{V^*(\mathbf{x}', t+\Delta, \eta)}\big]. \tag{12}$$

**Lemma 4** (Segment-level VRPO optimum). *Fix a state $(\mathbf{x}, t)$ and a short segment to $t+\Delta$. The VRPO optimum with temperature $\beta$ satisfies*

$$p_\theta(\mathbf{x}' \mid \mathbf{x}, t) = p_{\mathrm{ref}}(\mathbf{x}' \mid \mathbf{x}, t)\, \exp\Big(\tfrac{1}{\beta}V^*(\mathbf{x}', t+\Delta, 1/\beta) - \tfrac{1}{\beta}V^*(\mathbf{x}, t, 1/\beta)\Big). \tag{13}$$

*That is, the optimal conditional distribution is the Doob $h$-transform of the reference dynamics tilted by the terminal reward.*

*Proof.* The VRPO loss with temperature $\beta$ is

$$\mathcal{L}_{\mathrm{VRPO}} = -\mathbb{E}\left[\log \mathrm{sigmoid}\left(\beta \log \frac{p_\theta(\mathbf{x}'^+ \mid \mathbf{x}, t)}{p_{\mathrm{ref}}(\mathbf{x}'^+ \mid \mathbf{x}, t)} - \beta \log \frac{p_\theta(\mathbf{x}'^- \mid \mathbf{x}, t)}{p_{\mathrm{ref}}(\mathbf{x}'^- \mid \mathbf{x}, t)}\right)\right],$$

with preferences $\mathbf{x}'^+ \succ \mathbf{x}'^-$ iff $V^*(\mathbf{x}'^+, t+\Delta, 1/\beta) > V^*(\mathbf{x}'^-, t+\Delta, 1/\beta)$. Bradley–Terry analysis implies the optimal ratio has the form

$$\log \frac{p_\theta(\mathbf{x}' \mid \mathbf{x}, t)}{p_{\mathrm{ref}}(\mathbf{x}' \mid \mathbf{x}, t)} = C(\mathbf{x}, t) + \tfrac{1}{\beta}V^*(\mathbf{x}', t+\Delta, 1/\beta).$$

Normalizing over $\mathbf{x}'$ and using Equation (12), gives (13). □

**Lemma 5** (Inductive propagation of marginals). *Define the tilted marginal*

$$p^*(\mathbf{x}, t) \propto p_{\mathrm{ref}}(\mathbf{x}, t)\, \exp\big(\tfrac{1}{\beta}V^*(\mathbf{x}, t, 1/\beta)\big). \tag{14}$$

*If $p_\theta(\mathbf{x}, t) = p^*(\mathbf{x}, t)$ at time $t$, then applying the conditional kernel (13) yields $p_\theta(\mathbf{x}', t+\Delta) = p^*(\mathbf{x}', t+\Delta)$.*

*Proof.* Starting from $p_\theta(\mathbf{x}, t) = p^*(\mathbf{x}, t)$, propagate one segment:

$$p_\theta(\mathbf{x}', t+\Delta) = \int p_\theta(\mathbf{x}' \mid \mathbf{x}, t)\, p_\theta(\mathbf{x}, t)\, d\mathbf{x}$$

$$\propto \int p_{\mathrm{ref}}(\mathbf{x}' \mid \mathbf{x}, t)\, \exp\Big(\tfrac{1}{\beta}V^*(\mathbf{x}', t+\Delta, 1/\beta) - \tfrac{1}{\beta}V^*(\mathbf{x}, t, 1/\beta)\Big)$$

$$\cdot p_{\mathrm{ref}}(\mathbf{x}, t)\, \exp\big(\tfrac{1}{\beta}V^*(\mathbf{x}, t, 1/\beta)\big)\, d\mathbf{x}$$

$$= \exp\big(\tfrac{1}{\beta}V^*(\mathbf{x}', t+\Delta, 1/\beta)\big)\, p_{\mathrm{ref}}(\mathbf{x}', t+\Delta) \propto p^*(\mathbf{x}', t+\Delta).$$

Thus the tilted form is preserved. □

**Lemma 6** (Base case under independence (Domingo-Enrich et al., 2025)). *Suppose that under the reference process the initial and terminal states are independent, i.e., $p_{\mathrm{ref}}(\mathbf{x}_1 \mid \mathbf{x}_0, 0) = p_{\mathrm{ref}}(\mathbf{x}_1, 1)$. Then the optimal value satisfies*

$$V^*(\mathbf{x}_0, 0, \eta) = \log \mathbb{E}_{\mathbf{x}_1 \sim p_{\mathrm{ref}}(\cdot, 1)}\big[e^{\eta r(\mathbf{x}_1)}\big] =: C_\eta, \qquad V^*(\mathbf{x}_1, 1, \eta) = \eta\, r(\mathbf{x}_1).$$

*In particular, $V^*(\mathbf{x}_0, 0, \eta)$ is a constant (independent of $\mathbf{x}_0$).*

*Proof.* By definition of the value (closed form),

$$V^*(\mathbf{x}_0, 0, \eta) = \log \mathbb{E}_{\mathbf{x}_1 \sim p_{\text{ref}}(\cdot | \mathbf{x}_0, 0)}\big[e^{\eta r(\mathbf{x}_1)}\big].$$

Under the independence assumption, $p_{\text{ref}}(\mathbf{x}_1 \mid \mathbf{x}_0, 0) = p_{\text{ref}}(\mathbf{x}_1, 1)$, hence

$$V^*(\mathbf{x}_0, 0, \eta) = \log \mathbb{E}_{\mathbf{x}_1 \sim p_{\text{ref}}(\cdot, 1)}\big[e^{\eta r(\mathbf{x}_1)}\big] =: C_\eta,$$

which does not depend on $\mathbf{x}_0$. At the terminal time, the boundary condition gives $V^*(\mathbf{x}_1, 1, \eta) = \eta\, r(\mathbf{x}_1)$. $\qquad\square$

**Lemma 7** (Base case and terminal marginal). *At $t = 0$, the initial distribution of the learned process coincides with that of the reference process:*

$$p_\theta(\mathbf{x}, 0) \;=\; p_{\text{ref}}(\mathbf{x}, 0). \tag{15}$$

*Proof.* Since $\mathbf{x}_0$ and $\mathbf{x}_1$ are independent under the reference process, Lemma 6 implies

$$p_\theta(\mathbf{x}, 0) \;\propto\; p_{\text{ref}}(\mathbf{x}, 0) \, \exp(C_\eta),$$

where $C_\eta$ is a constant independent of $\mathbf{x}$. Normalization then yields (15). $\qquad\square$

**Theorem 3** (Optimality of VRPO). *Consider a trajectory on $\mathcal{T}_T$ that is decomposed into $K$ short segments, each of length $L$ with $\Delta = \frac{L}{T}$ sufficiently small. If VRPO is applied to segment-level preferences defined by $V_k^*(\mathbf{x}_t, t, \eta)$, then under sufficient model capacity, the resulting model is equivalent to training directly on the optimal target distribution*

$$p(\mathbf{x}_1) \;\propto\; p_{data}(\mathbf{x}_1) \, \exp\big(\eta\, r(\mathbf{x}_1)\big).$$

*Proof of Theorem 3.* Combining Lemmas 4, 5, and 7, we conclude that stage-wise VRPO training yields a terminal distribution

$$p(\mathbf{x}_1) \;\propto\; p_{\text{data}}(\mathbf{x}_1) \, \exp(\eta r(\mathbf{x}_1)),$$

which is exactly the optimal target. Moreover, it is straightforward to verify that the intermediate distributions also coincide. $\qquad\square$

Table 4: Results of quantitative comparison with baselines using the preference comparison prompts.

| Methods | PickScore | imageReward | Aesthetic score | HPSv2 |
|---------|-----------|-------------|-----------------|-------|
| SDXL | 22.2645 | 0.6738 | 6.4385 | 0.2786 |
| SPO | 23.3715 | 1.0234 | 6.8181 | 0.2909 |
| Ours | **23.6443** | **1.1823** | **7.2582** | **0.2916** |

## G   DETAILED COMPARISON OF TRAINING TIME

As shown in Table 5, compared to the standard DPO methods (taking DiffusionDPO as an example), VRPO adds an extra stage, but the time spent is significantly less than theirs. The purpose of the additional first stage is to train a step-wise value function, which allows VRPO to perform DPO at intermediate timesteps during diffusion image generation, rather than just after the image is fully generated. This approach saves a significant amount of time during the second stage of diffusion training.

Furthermore, we discovered that training just a prediction head of the step-wise value function yields very good results. We save the embeddings from before the prediction head, so the time required for this stage is much less than SPO.

Table 5: Comparison of training time between baseline methods.

| Methods/Stage | Value Function Training | Diffusion Training |
|---|---|---|
| Diffusion-DPO-SD1.5 | - | 384 A100 hours |
| SPO-SD1.5 | 92 A100 hours | 44 A100 hours |
| VRPO-SD1.5 | 0.3 A6000 hours | 62 A100 hours |
| Diffusion-DPO-SDXL | - | 4800 A100 hours |
| SPO-SDXL | 152 A100 hours | 124 A100 hours |
| VRPO-SDXL | 0.5 A6000 hours | 174 A100 hours |

Table 6: Results of ablation studies on segment length.

| Segment Length | PickScore | ImageReward | Aesthetic Score | HPSv2 |
|---|---|---|---|---|
| 1 | 22.8005 | 1.1912 | 6.7166 | 0.2805 |
| 2 | 22.8557 | 1.1925 | 7.0675 | 0.2828 |
| 3 | 22.8909 | **1.2698** | 7.0767 | 0.2829 |
| 4 | **23.0656** | 1.2404 | **7.1894** | 0.2867 |
| 6 | 23.0293 | 1.2208 | 7.0426 | **0.2875** |
| 8 | 22.9653 | 1.1933 | 6.8451 | 0.2850 |

## H MORE ABLATION STUDIES

**Segment Length**. We conduct experiments on different segment lengths, the results are shown in Table 6. As segment length increases from 1 to 4, performance consistently improves across nearly all metrics. This confirms that aggregating preference signal over multiple timesteps is highly beneficial. Beyond length 4, further increases yield diminishing returns and eventual degradation. Length 4 achieves the best trade-off and is used in all other experiments.

**Missranking**. Although precisely controlling the degree of missranking is challenging, we provide the following datapoints in Table 7: we trained multiple classification heads (50 classes) under different random seeds / training budgets, then evaluated them on a test split of 1,000 unseen trajectories. We report two losses from Algorithm 2 (CRPS and cross-entropy) as well as Top-1 and Top-3 accuracy of the predicted rank bucket.

## I DETAILED PROMPTS

Prompts for Figure 1 are provided in Table 8. Prompts for Figure 3 are provided in Table 9.

## J MLLM EVALUATION PROMPTS

The prompts used for evaluation with ChatGPT-5 were as follows:

Please evaluate and compare two images based on the following prompt and three dimensions:

Prompt associated with the images: {prompt}

Evaluation dimensions: 1. Overall performance: Comprehensive assessment of image quality, expressiveness and how well it fulfills the prompt 2. Visual appeal: Evaluation of the aesthetics, color matching and composition of the image 3. Text image alignment: Assessment of how well the image content matches the provided prompt

For each dimension, please clearly judge:

- Image 1 is better
- Image 2 is better

Table 7: Results of ablation studies on miss-ranking.

| Missranking | PickScore | ImageReward | Aesthetic Score | HPS v2 |
|---|---|---|---|---|
| CRPS=0.02293 CE=1.999 Top1-Acc:28.80% Top3-Acc:58.60% | 22.8824 | 1.2816 | 7.0424 | 0.2829 |
| CRPS=0.02070 CE=1.910 Top1-Acc:30.40% Top3-Acc:63.00% | 22.9602 | 1.1729 | 6.9367 | 0.2841 |
| CRPS=0.01631 CE=1.692 Top1-Acc:36.50% Top3-Acc:72.50% | 23.0656 | 1.2404 | 7.1894 | 0.2867 |

Table 8: Prompts for Figure 1.

| Row & Line | Prompt |
|---|---|
| Row 1, Line 1 | A beautiful natural woman. |
| Row 1, Line 3 | A drawing of a funny giraffe eating from a tree. |
| Row 1, Line 4 | Red and white high tops shoes and Bitcoin. |
| Row 1, Line 5 | A golden retriever representing god. |
| Row 1, Line 6 | Throne, dark scene, moonlight. |
| Row 1, Line 7 | Highway to hell that goes into a mountain covered in snow. |
| Row 2, Line 3 | Cute girl, Kyoto animation, 4k, high resolution. |
| Row 2, Line 4 | Corgi with helmet on bicycle. |
| Row 2, Line 5 | Fire sorcerer. |
| Row 2, Line 6 | A huskey playing footbal on the beach. |
| Row 3, Line 1 | Tripod legged robot in a lake. |
| Row 3, Line 2 | A robot that is made out of wood. |
| Row 3, Line 3 | Raccoon with a sombrero riding on a atv with Tequila in hand, photorealism. |
| Row 3, Line 4 | Hello kitty mecha, gears of war, style Artstation, octane render, unreal engine 6, epic game Graphics, Fantasy, cyberpunk, conceptual art, Ray tracing. |
| Row 3, Line 5 | A gloomy rabbit drinks wine. |

Table 9: Prompts for Figure 3.

| Line | Prompt |
|---|---|
| Line 1 | A beautiful Indian woman by the beach. |
| Line 2 | A man eating a glazed donut and a woman eating a chocolate cake. |
| Line 3 | Beautiful dancing unicorns. |
| Line 4 | A full-body shot of a beautiful woman wearing a dress, with sharp focus on her eyes, depicted in an elegant, intricate style by artgerm, jason chan, and mark hill. |
| Line 5 | Cowby angel standing in a field of flowers. |
| Line 6 | A cake with butterflies and rainbows. |
| Line 7 | A coloring book page of a lotus flower, white clear background. |
| Line 8 | A bee devouring the world. |

- Both are equal

Please answer in a clear and concise format, with each dimension on a separate line, for example:

- Overall performance: Image 1 is better

- Visual appeal: Both are equal

- Text image alignment: Image 2 is better

Although we allowed ChatGPT-5 to assign an "equal" score in the overall performance dimension, it never selected this option during evaluation.

