# OpenReview forum: "Value-as-Return: A Two-Stage Framework to Align on the Optimal Score Function"
_ICLR.cc/2026/Conference — Submitted to ICLR 2026_

### Official Review · Reviewer_dTqf · 2025-10-15

**Soundness:** 2
**Presentation:** 2
**Contribution:** 3
**Rating:** 4
**Confidence:** 3

**Summary:**

This paper addresses a critical and often-overlooked problem in the alignment of diffusion models via reinforcement learning. The authors identify a key limitation in existing methods based on Direct Preference Optimization, which incorrectly equate the likelihood of a full generation trajectory with the probability of the final output. The paper convincingly argues that this fundamental mismatch between trajectory-level processes and final-state rewards leads to suboptimal policy alignment and hinders the model's ability to learn complex preferences effectively.

**Strengths:**

The authors first formally articulate that existing DPO-based methods for diffusion models rely on a flawed oversimplification: they equate the intractable marginal probability of a final image with the joint probability of a single sampled trajectory. This leads to unstable training and suboptimal alignment, because the policy is updated using an incorrect and misleading reward signal. This is indeed a valid problem, and it would be beneficial to solve it efficiently without incurring high costs.

**Weaknesses:**

1. The paper does not discuss the practical computational cost of the proposed framework compared to baselines.
2. I hope the authors can share the hyperparameter sensitivity analysis regarding the segment length.
3. The transition from the continuous-time Stochastic Optimal Control (SOC) formulation in Section 2.2 to the discrete "bandit view" of DPO in Section 2.3 is abrupt. The writing can be improved.
4. The paper's theoretical conclusion points to a more direct alternative: guiding the diffusion process with the gradient of the reward function. This technique is well-established in the context of energy-based models and may represent a powerful baseline that can be included.

**Questions:**

See weaknesses.

---

> ### Author Response · Authors · 2025-11-24
> **Official Comment by Authors**
>
> # Response to Reviewer dTqf
> Dear Reviewer dTqf,
> Thank you very much for your careful reading and highly constructive feedback. In the following, we will address each of your comments in detail.
>
> > **W1: The paper does not discuss the practical computational cost of the proposed framework compared to baselines.**
>
> |Process|Diffusion-DPO-SD1.5|SPO-SD1.5|VRPO-SD1.5|Diffusion-DPO-SDXL|SPO-SDXL| VRPO-SDXL|
> |-|-|-|-|-|-|-|
> |Value Function Training|-|92 A100 hours|0.3 A6000 hours|-|152 A100 hours|0.5 A6000 hours|
> |Diffusion Training|384 A100 hours|44 A100 hours|62 A100 hours|4800 A100 hours|124 A100 hours|174 A100 hours|
>
> Compared to the standard DPO methods (taking DiffusionDPO as an example), VRPO adds an extra stage, but the time spent is **significantly less** than theirs. The purpose of the additional first stage is to train a step-wise value function, which allows VRPO to perform DPO at intermediate timesteps during diffusion image generation, rather than just after the image is fully generated. This approach saves a significant amount of time during the second stage of diffusion training.
>
> Furthermore, we discovered that training just a prediction head of the step-wise value function yields very good results. We save the embeddings from before the prediction head, so the time required for this stage is much less than SPO and can be considered almost negligible.
>
> In addition to its **state-of-the-art performance** and **minimal computing resource** requirements during training, we will also **open-source** the rollout images, training and inference codes, and checkpoints to facilitate future research.
>
> > **W2. I hope the authors can share the hyperparameter sensitivity analysis regarding the segment length.**
>
> Thank you for the suggestion. We will include this ablation study on the segment length hyperparameter in the revision. Results are reported below:
> | Segment Length | PickScore     | ImageReward     | Aesthetic Score     | HPS v2    |
> |---------------|--------------|--------------|--------------|--------------|
> | 1     | 22.8005      | 1.1912       | 6.7166       | 0.2805       |
> | 2     | 22.8557      | 1.1925       | 7.0675       | 0.2828       |
> | 3     | 22.8909      | **1.2698**   | *7.0767*        | 0.2829       |
> | 4     | **23.0656**  | *1.2404*   | **7.1894**   | *0.2867*    |
> | 6     | *23.0293*   |   1.2208      | 7.0426  | **0.2875**  |
> | 8     | 22.9653   | 1.1933    | 6.8451 | 0.2850   |
>
> As segment length increases from 1 to 4, performance consistently improves across nearly all metrics. This confirms that aggregating preference signal over multiple timesteps is highly beneficial. Beyond length 4, further increases yield diminishing returns and eventual degradation. Length 4 achieves the best trade-off and is used in all other experiments.
>
> > **W3: The transition from the continuous-time Stochastic Optimal Control (SOC) formulation in Section 2.2 to the discrete "bandit view" of DPO in Section 2.3 is abrupt. The writing can be improved.**
>
> Thank you for your insightful comment. We acknowledge the transition from Section 2.2 to Section 2.3 could be smoother. We will make sure to include additional transitional remarks in the final version to better clarify the relationship between these two sections.
>
> > **W4: The paper's theoretical conclusion points to a more direct alternative: guiding the diffusion process with the gradient of the reward function. This technique is well-established in the context of energy-based models and may represent a powerful baseline that can be included.**
>
> Thank you for your valuable suggestion. We will include this baseline in the revision. We tested using our best value function as classifier guidance, incorporating the gradient from the value function in Lemma 1 to guide the diffusion denoising process. The results are shown below:
> | Method        | PickScore     | ImageReward     | Aesthetic Score     | HPS v2    |
> |---------------|--------------|--------------|--------------|--------------|
> |SDXL|21.7093|0.5887|6.4658|0.2738|
> |Classifier Guidance|21.7768|0.6022|6.4716|0.2738|
> |MAPO|21.9038|0.7429|6.5918|0.2791|
> |Diffusion-DPO|22.3269|0.9411|6.3959|0.2837|
> | SPO           | 22.7821      | 1.0536       | 6.8035       | 0.2858       |
> | VRPO (Previous) | 22.9771 | 1.1523 | 6.8783 | 0.2847
> | VRPO (Revision)    | **23.0656**  | **1.2404**   | **7.1894**   | **0.2867**   |
>
> Despite extensive hyper-parameter tuning, we could not achieve particularly better results. If the classifier guidance weight is too small, there is little to no change; if it's too large, we observe artifacts and a decrease in evaluation metrics. Here are some datapoints:
> |Classifier Guidance Scale|PickScore|ImageReward|Aesthetic Score|HPS v2|
> |-|-|-|-|-|
> |0.0 (SDXL)|21.7093|0.5887|6.4658|0.2738|
> |8.0|21.7353|0.5798|6.5223|0.2736|
> |16.0|21.7721|0.5752|6.5023|0.2735|
> |32.0|21.7768|0.6022|6.4716|0.2738|
> |64.0| 21.4073|0.5661|6.4579|0.2735|

---

### Official Review · Reviewer_CTuf · 2025-10-29

**Soundness:** 3
**Presentation:** 3
**Contribution:** 2
**Rating:** 4
**Confidence:** 2

**Summary:**

This paper identifies a fundamental limitation in existing Direct Preference Optimization (DPO) methods for diffusion models: the oversimplification of equating the probability of a full generation trajectory with the marginal probability of the final image. To address this problem, the authors propose a two-stage reinforcement learning framework: First, a value-distribution function is learned to estimate the distribution of returns (future rewards) from any intermediate state. Second, a novel Value-as-Return Preference Optimization (VRPO) method is applied to short trajectory segments, using the learned value function to label preferences. The paper proves that under sufficient model capacity, their method converges to an optimal policy that samples from a "tilted" target distribution, and experiments on large-scale diffusion models demonstrate that VRPO achieves some improvements over several strong baselines.

**Strengths:**

To the best of my knowledge, the proposed method is novel and addresses a key shortcoming of prior work, where the probability of full generation trajectories are treated as the marginal probability of the final image. The writing is also generally clear and easy to follow, and the authors evaluate their method on large scale models relevant to practitioners working with diffusion models. Additionally, the approach is grounded theoretically.

**Weaknesses:**

My main concern revolves around the significance of the author’s empirical results. To me it looks like VRPO marginally outperformed SPO on SDXL (Table 1) with respect to the metrics considered, except for ImageReward where gains are more significant. I am having difficulty gaging why the authors claim that these are substantial improvements over the SPO baseline; have similar gains by other methods (e.g., 1-3% improvements) been considered significant in the past? Or is the ImageReward a metric readers might care about more than the others?

**Questions:**

Can the authors elaborate on why their main empirical results in Table 1 are meaningful improvements over the strongest baseline, SPO?

---

> ### Author Response · Authors · 2025-11-24
> **Official Comment by Authors**
>
> # Response to Reviewer CTuf
> Dear Reviewer CTuf,
> Thank you very much for your careful reading and highly constructive feedback. In the following, we will address each of your comments in detail.
>
> > **W1 & Q1: My main concern revolves around the significance of the author’s empirical results. To me it looks like VRPO marginally outperformed SPO on SDXL (Table 1) with respect to the metrics considered, except for ImageReward where gains are more significant. I am having difficulty gaging why the authors claim that these are substantial improvements over the SPO baseline; have similar gains by other methods (e.g., 1-3% improvements) been considered significant in the past? Or is the ImageReward a metric readers might care about more than the others? Can the authors elaborate on why their main empirical results in Table 1 are meaningful improvements over the strongest baseline, SPO?**
>
> We believe VRPO achieves substantial and meaningful progress for the following reasons:
>
> (1) VRPO’s advantage on PickScore is actually significant. PickScore is a pairwise preference model trained on image pairs using cross-entropy loss on softmaxed logits. The reported values are averages of these logit values, which are centered far from zero (~18–23). As a result, relative differences do not scale linearly with preference strength, and percentage improvements are not a reliable measure of practical advantage.
>
> A more informative signal is the pairwise win rate: VRPO defeats SPO in 63.82% (now 69.72% in the revision) of head-to-head comparisons. Moreover, when VRPO wins, the PickScore margin is typically large, while losses occur by small margins. This asymmetry is clearly visible in Figure 2 (left), where VRPO shows a pronounced rightward shift and strongly dominates SPO in the high-quality regime (>23.0).
>
> (2) Qualitative results strongly favor VRPO. As illustrated in Figure 3, across a wide range of prompts, VRPO generations consistently exhibit superior composition, prompt fidelity, and overall detail richness.
>
> (3) Explanation of the relative importance of evaluation metrics. PickScore is our primary optimization target and thus most directly reflects algorithmic effectiveness, whereas metrics such as ImageReward primarily evaluate generalization, an equally critical aspect of RL-based alignment. We are glad the reviewer acknowledges the large gain on ImageReward; combined with the substantial pairwise advantage on PickScore described in (1), this reinforces that VRPO represents a meaningful advance.
>
> Finally, following further training of our VRPO-SDXL model, we have updated the main results in the revised manuscript using our latest checkpoint (publicly available on the anonymous website). The new numbers show considerably larger margins than previously reported:
> | Method|PickScore|ImageReward|Aesthetic Score|HPS v2|
> |-|-|-|-|-|
> |SPO|22.7821|1.0536|6.8035|0.2858|
> |VRPO (Previous)|22.9771|1.1523|6.8783|0.2847|
> |VRPO (Revision)|**23.0656**| **1.2404**|**7.1894**|**0.2867**|

---

### Official Review · Reviewer_xTqE · 2025-11-03

**Soundness:** 4
**Presentation:** 3
**Contribution:** 2
**Rating:** 4
**Confidence:** 3

**Summary:**

The paper is presenting an algorithm for modifying a diffusion model’s induced distribution to align it with a specified reward function, and in particular learning a generative model for the tilted distribution $p(x) e^{r(x)}$. The approach is split into two stages: first learn a value or cumulative distribution over short diffusion segments, then fine-tune the diffusion model by ranking segment endpoints using that learned value.

**Strengths:**

- The theory section builds from reasonable assumptions and gives good theoretical intuition of the algorithm.
- The method is easy to follow and to implement.
- Results show some practical improvements on metrics on Stable Diffusion 1.5. While not uniformly large, the gains are nontrivial in places and indicate the approach is at least competitive.

**Weaknesses:**

- The paper does not position itself clearly against [Adjoint Matching](https://arxiv.org/abs/2409.08861), which is a closely related framework that also targets a tilted terminal distribution. Similar to e.g. [Diffusion-QL](https://arxiv.org/abs/2208.06193) and e.g. [QSM](https://arxiv.org/abs/2312.11752) for diffusion RL, there are both pros and cons for using a value-function-based vs. vector field-based approach. Without any comparisons, it is hard to know when a practitioner should choose this method instead of something like adjoint matching.

- While results on SD 1.5 are nonnegligible, the improvements on SDXL seem very marginal. When there are many different methods for learning a tilted distribution for score-based generative models, it makes it further unclear as to when this method should be used vs. alternatives.
- The paper does not quantify compute cost or runtime relative to strong baselines. Likewise to the above points, lack of this information makes it difficult for future researchers to evaluate when they would want to use this method.

**Questions:**

- Under what conditions should a practitioner prefer this method over Adjoint Matching?
- How sensitive is performance to segment length, to misranking in the value model over time steps, and to the inverse temperature?
- What are the compute and runtime characteristics compared with other baselines?

---

> ### Author Response · Authors · 2025-11-24
> **Official Comment by Authors (Part One)**
>
> Dear Reviewer xTqE,
> Thank you very much for your careful reading and highly constructive feedback. In the following, we will address each of your comments in detail.
>
> > ### **An Overall Response**
> We recognize that your insightful focus lies in evaluating whether VRPO should be selected among various algorithms for diffusion RL.
> Your clear and rigorous breakdown into three decisive dimensions: (W1) the theoretical grounding and conceptual rationale, (W2) the experimental performance, and (W3) the computational resource requirements, offers an excellent framework for evaluation. Following this structure, we detail below why VRPO is a highly competitive method and a compelling direction for future work.
>
> > **W1 & Q1: The paper does not position itself clearly against Adjoint Matching. Under what conditions should a practitioner prefer this method over Adjoint Matching?**
>
> We thank you very much for pointing this out. We acknowledge that this point was not clearly articulated earlier, and it is crucial to emphasize the following distinctions in the final Discussion section. We also welcome any additional feedback.
>
> **Summary Closed-Form Analysis**
> - **Adjoint Matching** is the *first* method to provide a closed-form analysis in a **vector-based** framework, using the quantity \\(\partial r / \partial X_t\\) to guide updates of \\(v\\).
> - **Our method** is the *first* to provide a closed-form analysis in a **value-based** framework.
>
> **Limitations of Adjoint Matching**
> - Builds on ReFL and requires computing gradients from the reward signal back to the noisy image.
> - Relies on Jacobian–vector products (JVPs) to backpropagate through the entire \\(v\\)-prediction network. Specifically, as shown in Equation (41) of the arXiv version of Adjoint Matching, it first requires computing \\(\partial r / \partial X_1\\), then taking a dot product with \\(2v^{\text{base}}(X_t, t) - \frac{\dot{\alpha}_t}{\alpha_t} X_t\\), and finally computing the partial derivative of this resulting expression with respect to \\(X_t\\).
> - Must **iteratively compute these gradients across timesteps**, leading to:
>
>     - High GPU memory consumption
>     - Slow computation, especially when v-prediction net is very large
>     - Accumulated approximation error over the diffusion trajectory
>
> - This setup diverges from classical RL assumptions, where reward-to-state gradients are typically inaccessible, though this limitation may be less concerning in the context of diffusion models.
>
>  **Advantages of Our Method**
> - Follows the **DPO framework** and is guided by **ROC theory**.
> - Provides the **first value-function-based approach that remains consistent across timesteps**.
> - Avoids backward-through-generation operations, preventing error accumulation and reducing computational cost.
> - Naturally supports **Monte Carlo estimation**, enabling applicability even without a trained reward model.
>
>
> > **W2: While results on SD 1.5 are nonnegligible, the improvements on SDXL seem very marginal. When there are many different methods for learning a tilted distribution for score-based generative models, it makes it further unclear as to when this method should be used vs. alternatives.**
>
> We will update the metrics with the latest VRPO-SDXL checkpoint (available at the anonymous site) in the revision, as shown in the table below.
>
> | Method|PickScore|ImageReward|Aesthetic Score|HPS v2|
> |-|-|-|-|-|
> |SPO|22.7821|1.0536|6.8035|0.2858|
> |VRPO (Previous)|22.9771|1.1523|6.8783|0.2847|
> |VRPO (Revision)|**23.0656**| **1.2404**|**7.1894**|**0.2867**|
>
> A more informative signal is the pairwise win rate: VRPO (Revision) defeats SPO in **69.72%** of head-to-head PickScore comparisons.
>
> > **W3 & Q3: The paper does not quantify compute cost or runtime relative to strong baselines. Likewise to the above points, lack of this information makes it difficult for future researchers to evaluate when they would want to use this method.**
>
> Here is the comparison of *Computing cost:*
> |Process|Diffusion-DPO-SD1.5|SPO-SD1.5|VRPO-SD1.5|Diffusion-DPO-SDXL|SPO-SDXL| VRPO-SDXL|
> |-|-|-|-|-|-|-|
> |Value Function Training|-|92 A100 hours|0.3 A6000 hours|-|152 A100 hours|0.5 A6000 hours|
> |Diffusion Training|384 A100 hours|44 A100 hours|62 A100 hours|4800 A100 hours|124 A100 hours|174 A100 hours|
>
> *Inference runtime:*
> All compared methods produce either LoRA adapters or full-parameter checkpoints that are applied to the same base diffusion UNet. Therefore, once the checkpoint is loaded, inference speed and memory footprint are identical to the base diffusion model across all methods.
>
> In addition to its **state-of-the-art performance** and **minimal computing resource** requirements during training, we will also **open-source** the rollout images, training and inference codes, and checkpoints to facilitate future research. We believe this will make VRPO an attractive choice for future researchers in the field.

---

> ### Author Response · Authors · 2025-11-24
> **Official Comment by Authors (Part Two)**
>
> > **Q2. How sensitive is performance to segment length, to misranking in the value model over time steps, and to the inverse temperature?**
>
> Thank you for your valuable suggestion. We will include these ablation studies in the revision.
>
> **Segment length**: We have conducted an ablation on the segment length hyperparameter. Results are reported below:
>
> |Segment Length|PickScore|ImageReward|Aesthetic Score|HPS v2|
> |-|-|-|-|-|
> |1|22.8005|1.1912|6.7166|0.2805|
> |2|22.8557|1.1925|7.0675|0.2828|
> |3|22.8909|**1.2698**|*7.0767*|0.2829|
> |4|**23.0656**|*1.2404*|**7.1894**|*0.2867*|
> |6|*23.0293*|1.2208|7.0426|**0.2875**|
> |8|22.9653|1.1933|6.8451|0.2850|
>
> As segment length increases from 1 to 4, performance consistently improves across nearly all metrics. This confirms that aggregating preference signal over multiple timesteps is highly beneficial. Beyond length 4, further increases yield diminishing returns and eventual degradation. Length 4 achieves the best trade-off and is used in all other experiments.
>
> **Missranking**: Although precisely controlling the degree of missranking is challenging, we provide the following datapoints: we trained multiple classification heads (50 classes) under different random seeds / training budgets, then evaluated them on a test split of 1,000 unseen trajectories. We report two losses from Algorithm 2 (CRPS and cross-entropy) as well as Top-1 and Top-3 accuracy of the predicted rank bucket.
>
> |Missranking|PickScore|ImageReward|Aesthetic Score|HPS v2|
> |-|-|-|-|-|
> |CRPS=0.02293 CE=1.999 Top1-Acc:28.80% Top3-Acc:58.60%|22.8824|**1.2816**|*7.0424*|0.2829|
> |CRPS=0.02070 CE=1.910 Top1-Acc:30.40% Top3-Acc:63.00%|22.9602|1.1729|6.9367|0.2841|
> |CRPS=0.01631 CE=1.692 Top1-Acc:36.50% Top3-Acc:72.50%|**23.0656**|*1.2404*|**7.1894**|**0.2867**|
>
> PickScore, serving as the training objective, exhibits a clear positive correlation with value-model ranking accuracy: better ranking → significantly higher PickScore. Generalization metrics (ImageReward, Aesthetic Score, HPS v2) show weaker correlation.
>
> **Inverse temperature:** In our experiments, the coefficient η in Lemma 1 has limited impact on final performance when kept in a reasonable range around 1/β, consistent with maintaining the inverse proportionality derived theoretically. We observed only minor variations in metrics across η ∈ [0.25/β, 4/β], confirming the robustness predicted by the lemma.

---

### Official Review · Reviewer_ZhML · 2025-11-11

**Soundness:** 3
**Presentation:** 3
**Contribution:** 3
**Rating:** 6
**Confidence:** 2

**Summary:**

introduces Value-as-Return (VRPO), a two-stage reinforcement learning framework for aligning diffusion models with human preferences. It aims to address a flaw in prior DPO methods that misinterpret trajectory and final-state probabilities. VRPO learns a value-distribution function for short segments, then refines the model using these learned returns. the authors report more stable and a consistently superior alignment method for large-scale diffusion models.

**Strengths:**

- provides a theoretically grounded solution using stochastic optimal control, correcting a major flaw in existing DPO-based diffusion training
- the value-distribution + VRPO pipeline improves reward consistency and stability across long diffusion trajectories.
- demonstrates clear, consistent gains over strong baselines (e.g., SPO, Diffusion-DPO) on large-scale diffusion models like SDXL.

**Weaknesses:**

- the two-stage training process (learning a value-distribution function before VRPO) adds computational and implementation overhead compared to simpler DPO methods.
- experiments focus mainly on image diffusion models, leaving it unclear how well the approach generalizes to other domains or more diverse reward settings. It would be interesting to compare this to conventional sequential decision-making tasks

**Questions:**

- do the authors plan to evaluate their method on sequential decision-making tasks? e.g. robotics, atari, and continuous control settings.

---

> ### Author Response · Authors · 2025-11-24
> **Official Comment by Authors**
>
> # Response to Reviewer ZhML
> Dear Reviewer ZhML,
> Thank you very much for your careful reading and highly constructive feedback. In the following, we will address each of your comments in detail.
>
> > **W1: The two-stage training process (learning a value-distribution function before VRPO) adds computational and implementation overhead compared to simpler DPO methods.**
>
> Yes, compared to the standard DPO methods (taking DiffusionDPO as an example), VRPO adds an extra stage, but the time spent is **significantly less** than theirs. The purpose of the additional first stage is to train a step-wise value function, which allows VRPO to perform DPO at intermediate timesteps during diffusion image generation, rather than just after the image is fully generated. This approach saves a significant amount of time during the second stage of diffusion training.
>
> Furthermore, we discovered that training just a prediction head of the step-wise value function yields very good results. We save the embeddings from before the prediction head, so the time required for this stage is much less than SPO and can be considered almost negligible.
>
> |Process|Diffusion-DPO-SD1.5|SPO-SD1.5|VRPO-SD1.5|Diffusion-DPO-SDXL|SPO-SDXL| VRPO-SDXL|
> |-|-|-|-|-|-|-|
> |Value Function Training|-|92 A100 hours|0.3 A6000 hours|-|152 A100 hours|0.5 A6000 hours|
> |Diffusion Training|384 A100 hours|44 A100 hours|62 A100 hours|4800 A100 hours|124 A100 hours|174 A100 hours|
>
> > **W2 & Q1: Experiments focus mainly on image diffusion models, leaving it unclear how well the approach generalizes to other domains or more diverse reward settings. It would be interesting to compare this to conventional sequential decision-making tasks. Do the authors plan to evaluate their method on sequential decision-making tasks? e.g. robotics, atari, and continuous control settings.**
>
> We appreciate the reviewer’s insightful suggestion. However, our team's expertise is primarily focused on generative 2D diffusion models, and we currently lack the domain knowledge and resources necessary to conduct rigorous experiments in sequential decision-making tasks such as robotics, Atari, or continuous control. As such, we believe that extending VRPO to these areas is best suited for future research by those with relevant expertise.

---

### Author Response · Authors · 2025-11-24
**Rebuttal to all reviewers**

We sincerely thank all the reviewers for their invaluable feedback and contributions, and we apologize for the delayed response.

- We extend our heartfelt thanks to reviewer ZhML for their recognition of the theoretical grounding of VRPO and its consistent empirical gains over strong baselines. **Their feedback enabled us to more clearly and thoroughly demonstrate the computational efficiency advantages of our method**, thereby making the paper clearer and more convincing.
- We are deeply grateful to reviewer xTqE for their strong recognition of the theoretical soundness of our work. **Their valuable suggestion to compare against a broader range of RL algorithms greatly broadened the scope of our study**, allowing it to reach a wider audience.
- We sincerely thank reviewer CTuf for recognizing the novelty of identifying and addressing the flaws in prior diffusion DPO methods, as well as the strong theoretical grounding and clear presentation of our work. **Their thoughtful suggestion to clarify the evaluation metrics significantly enhanced the clarity, comprehensibility, and overall completeness of the paper**, thereby improving the accessibility of our contributions.
- We extend our heartfelt gratitude to reviewer dTqf for acknowledging that the core problem we identified is both critical and frequently overlooked, and for highlighting that an efficient solution would meaningfully benefit the community. **Their insightful feedback motivated us to enrich the paper with additional details on computational cost, ablation studies, writing, and baseline comparisons**, which markedly strengthened the persuasiveness, rigor, and overall robustness of our approach.

---

### Meta-Review · Area_Chair_Q9jh · 2026-01-05

**Summary:**

This paper proposes a framework aimed at addressing the DPO mismatch problem in diffusion tasks. While the reviewers acknowledge the motivation behind this work, the empirical evaluation does not convincingly demonstrate the effectiveness of the proposed VRPO framework. Specifically, the performance gains over baseline methods such as SPO and SPXL appear marginal and lack statistical significance. Consequently, I recommend rejection in its current form, and encourage the authors to strengthen the experimental validation for future submission.

**Reviewer Concerns:**

The major remaining concern is the empirical performance of the proposed VRPO framework. First, its overall gains over strong baselines like SPXL are not significant. Second, the two-stage design introduces non-trivial computational and implementation overhead compared to simpler DPO alternatives, without a commensurate performance advantage.

**Reviewer Scores:**

I think the reviewers would have kept their scores if fully engagement in the discussion was possible.

---

### Decision · Program_Chairs · 2026-01-26

Reject